# The valley Nernst effect in WSe$_2$

Minh Tuan Dau[1]*, Céline Vergnaud[1], Alain Marty [1], Cyrille Beigné[1], Serge Gambarelli[2], Vincent Maurel[2], Timotée Journot[3], Bérangère Hyot[3], Thomas Guillet[1], Benjamin Grévin[2], Hanako Okuno[4] & Matthieu Jamet[1]*

The Hall effect can be extended by inducing a temperature gradient in lieu of electric field that is known as the Nernst (-Ettingshausen) effect. The recently discovered spin Nernst effect in heavy metals continues to enrich the picture of Nernst effect-related phenomena. However, the collection would not be complete without mentioning the valley degree of freedom benchmarked by the valley Hall effect. Here we show the experimental evidence of its missing counterpart, the valley Nernst effect. Using millimeter-sized WSe$_2$ mono-multi-layers and the ferromagnetic resonance-spin pumping technique, we are able to apply a temperature gradient by off-centering the sample in the radio frequency cavity and address a single valley through spin-valley coupling. The combination of a temperature gradient and the valley polarization leads to the valley Nernst effect in WSe$_2$ that we detect electrically at room temperature. The valley Nernst coefficient is in good agreement with the predicted value.

[1] Univ. Grenoble Alpes, CEA, CNRS, Grenoble INP, IRIG-SPINTEC, 38000 Grenoble, France. [2] Univ. Grenoble Alpes, CEA, CNRS, IRIG-SyMMES, 38000 Grenoble, France. [3] Univ. Grenoble Alpes, CEA, LETI, MINATEC Campus, 38000 Grenoble, France. [4] Univ. Grenoble Alpes, CEA, IRIG-MEM, 38000 Grenoble, France. *email: dautuan@gmail.com; matthieu.jamet@cea.fr

Semiconducting transition metal dichalcogenides (TMDCs) have gained a great attention because of their fascinating electronic and optical properties when downing the thickness to one monolayer[1,2]. Many of the TMDCs have since been shown to exhibit a strong excitonic feature, showing strong and spectrally narrow photoluminescence and absorption making them very good candidates for optoelectronic devices[3,4]. The combination of a bandgap (1–2 eV) and an atomically thin conducting channel also makes TMDCs ideal materials for next generation field effect transitors in microelectronics[5]. Finally, in the monolayer form, due to the breaking of inversion symmetry and strong spin–orbit coupling, the carriers in TMDCs carry a valley index noted $K^+$ or $K^-$. This valley degree of freedom refers to local extrema in the band structure of TMDC monolayers at the $K$ points of the two-dimensional Brillouin zone and it can be exploited to develop novel valleytronic devices[6–8]. In $MX_2$ (M = Mo, W and X = S, Se) monolayers, the broken inversion symmetry and strong spin–orbit coupling originating from the $d$-orbitals of the transition metal atom align the electron spins near opposite valleys $K^+$ and $K^-$ to opposite out-of-plane directions thus preserving time reversal symmetry. The broken spatial inversion symmetry also generates large Berry curvatures near the $K$ valleys, the Berry curvatures being opposite at opposite valleys[9]. This effect gives rise to unconventional electronic properties such as the valley Hall effect[6,10]. An interesting growing field of investigation consists in coupling the valley physics with thermoelectrics. Several spin/valley-dependent thermoelectric effects have already been predicted theoretically in monolayer TMDCs such as the spin Nernst effect (thermoelectric counterpart of the spin Hall effect[11–13]) and the valley Nernst effect (VNE)[14,15]. Hence, the experimental demonstration of those effects could give rise to a new field called valley caloritronics (by analogy with spin caloritronics[16]) with the advantage that the coupled valley-spin degree of freedom is more robust than the spin with respect to external magnetic fields due to the large intrinsic spin–orbit coupling. One main obstacle to the observation of the VNE comes from the absence of large area TMDC monolayers which is a prerequisite to apply macroscopic temperature gradients. Here we use the van der Waals epitaxy to grow high-quality $WSe_2$ mono- and multilayer on epitaxial graphene on SiC over 1 cm$^2$[17,18]. By this, we are able to apply temperature gradients in $WSe_2$ on mm scale. However, in order to detect electrically the VNE, one needs to lift the $K^+$–$K^-$ valley degeneracy, i.e., breaking time reversal symmetry and creating a population imbalance between the two valleys. To do so, we use the ferromagnetic resonance-spin pumping (FMR-SP) technique[19–22] by growing in-situ the Al/NiFe/Al magnetic stack on top of the $WSe_2$ layer. By suitably adjusting the position of the sample into the cavity, at the FMR of NiFe ($Ni_{81}Fe_{19}$), we are able to: (i) generate a temperature gradient along the sample and (ii) address a single valley by spin pumping when the magnetic field is applied perpendicular to the film. Eventually, we detect the VNE which is in quantitative agreement with theoretical predictions.

## Results

### Growth of $WSe_2$ and characterization.
The growth of mono- and multilayer of $WSe_2$ is performed under ultrahigh vacuum by molecular beam epitaxy (MBE) in the van der Waals regime on epitaxial graphene on SiC. The graphene layer was grown by chemical vapor deposition on the Si face of SiC. The VNE is expected to be very large and dominant at the top of the valence band at the $K^+$ and $K^-$ points of the 2D Brillouin zone. Hence $WSe_2$ layers are $p$ doped by introducing 0.1% of Nb in order to shift the Fermi level to the top of the valence band[23]. In Fig. 1b, ex-situ grazing incidence x-ray diffraction shows the single crystalline character of the $WSe_2$ layers and the measured lattice

parameter which is the one of bulk $WSe_2$. More characterization results on the $WSe_2$ layers are given in the Supplementary Note 1. We demonstrate that the Nb incorporation does not affect the growth and crystallinity of the $WSe_2$ layers but it makes them conducting at room temperature and provides electronic states for carrier diffusion and the detection of the VNE (see Supplementary Note 4). $WSe_2$ films are finally capped with a thin ≈10-nm-thick amorphous Se layer to allow air-transfer to another deposition chamber to grow the Al(7 nm)/NiFe(20 nm)/Al(5 nm) stack after the thermal desorption of the amorphous Se.

Figure 1c shows the cross-sectional scanning transmission electron microscopy (STEM) micrographs with high-angle annular dark-field imaging (HAADF) technique of the stacks made of monolayer $WSe_2$ (Fig. 1c (top left) and 1c (bottom left)) and multilayer $WSe_2$ (Fig. 1c (top right) and 1c (bottom right)) used for the VNE experiments. The interfaces (graphene/$WSe_2$) and ($WSe_2$/metallic stack) with high crystalline quality are atomically resolved and abrupt. The number of $WSe_2$ layers can be unambiguously determined based on the STEM images, and is in agreement with the estimation done by the combination of atomic force microscopy measurements and quartz crystal microbalance. We clearly note that the uppermost layer $WSe_2$ is degraded after Al deposition on top which is likely due to an intermixing of Al and Se. For that reason, in order to preserve the desired number of layers, especially in the case of one monolayer $WSe_2$, we have systematically deposited at least 30% of matter (Se, W, Nb) more.

Figure 2 illustrates the temperature-gradient induced voltages that are at play under the application of an external magnetic field coupled with spin injection via FMR-spin pumping. We also include rectification effects (RE) due to the anisotropic magnetoresistance and anomalous Hall effect in NiFe. They do not contribute to the measured voltage when the magnetic field and the magnetization are perpendicular to the film (i.e., $\theta_H = 90°$ and 270°)[20,24]. It is shown that the VNE might be detected in the configuration where the magnetic field is perpendicular to the sample surface. We discuss below how we can electrically detect the VNE thanks to the FMR-SP technique and disqualify the spurious effects based on robust arguments.

### FMR-spin pumping experiments.
In order to study the VNE, the samples are cut into small $0.4 \times 2.4$ mm$^2$ pieces, electrically contacted with Al wires using a mechanical bonding machine and finally introduced into the 9 GHz electron paramagnetic resonance cavity (Fig. 3a). The radio frequency (RF) power is set to 200 mW. In FMR-SP experiments, thermoelectric effects like the ordinary Seebeck effect (OSE) are considered as parasitic effects and are usually removed from electrical signals[25]. Here, we purposely position the sample away from the center of the cavity in order to generate a temperature gradient along the sample and study the VNE as shown in Fig. 3a. By shifting up (resp. down) the sample by $\Delta y$, microwaves are more absorbed by the NiFe layer at the bottom (resp. top) of the sample with respect to the top (resp. bottom) leading to a temperature gradient along $-\hat{y}$ (resp. $+\hat{y}$). We measure simultaneously the FMR signal of the NiFe layer (Fig. 3b, c) and the voltage $V = V_1 + V_t$ as a function of the angle $\theta_H$ from 0° to 360°. The DC magnetic field $H_{static}$ has a fixed direction along $+\hat{x}$. $V_1$ (resp. $V_t$) corresponds to the longitudinal (resp. transverse) component of the voltage. When rotating the sample, we find that the magnetization of the soft NiFe layer is parallel to the applied field at $\theta_H = 90°$ and 270° at the ferromagnetic resonance by using a simple macrospin model and by minimizing the system energy[20]. We confirmed this prediction by out-of-plane anisotropic magnetoresistance measurements (see Supplementary Note 3).

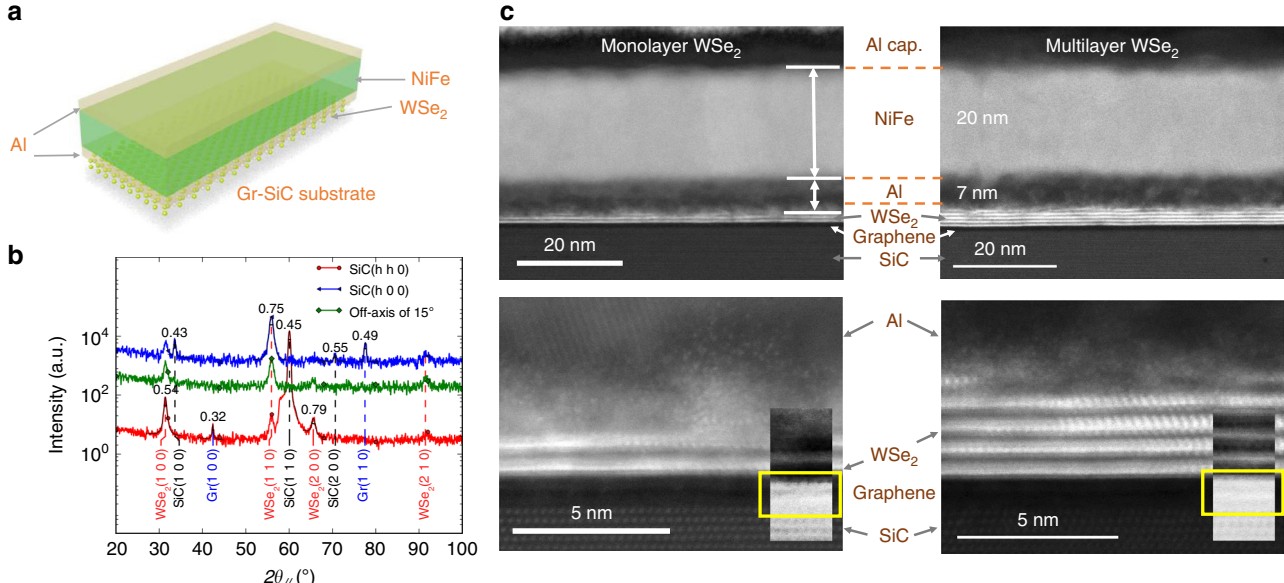

**Fig. 1 Characterizations of the WSe$_2$ layers. a** Schematic of the stack grown in-situ and employed for the VNE experiments. **b** In-plane $\theta/2\theta$ scans along different crystallographic axis with respect to the SiC substrate. Full width at half maximum of reflection peaks are indicated. **c** STEM-HAADF images of the two stacks composed of one monolayer WSe$_2$ (top left) and multilayer WSe$_2$ (top right). Abrupt interfaces between WSe$_2$ and graphene of the stacks can be identified at atomic scale when increasing the magnification: monolayer WSe$_2$ (bottom left) and multilayer WSe$_2$ (bottom right). The insets are bright-field images showing graphene sheets (yellow box).

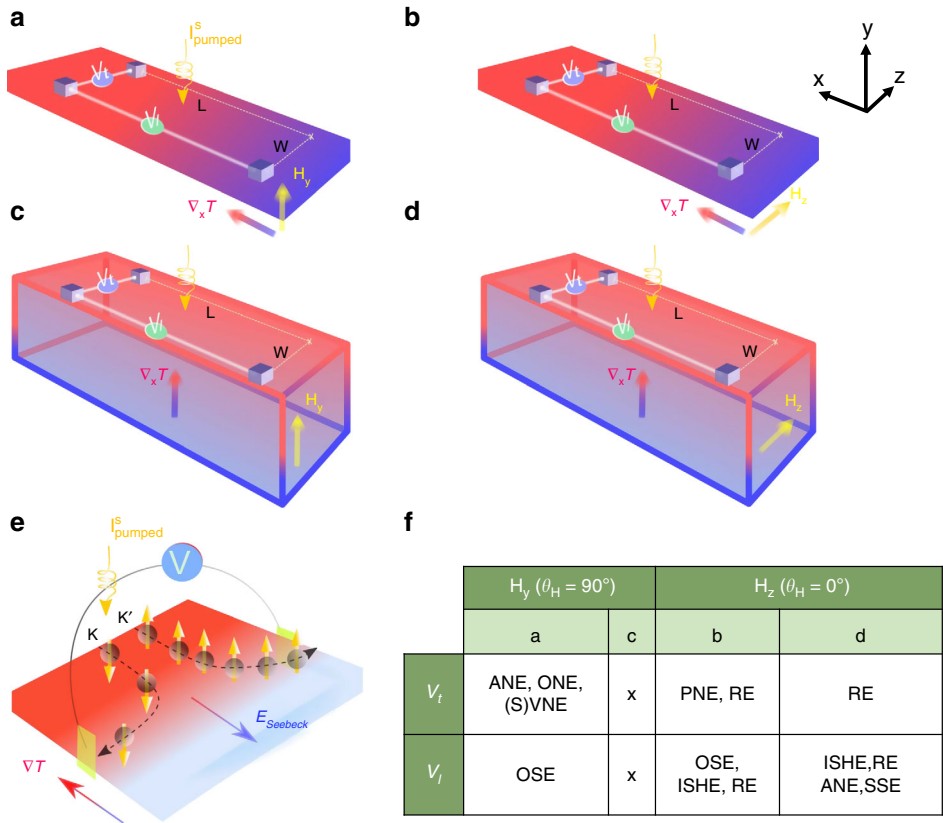

**Fig. 2 Schematics of the transverse ($V_t$), longitudinal ($V_l$) voltages measured under different configurations and the electrical detection of the VNE.** The external applied magnetic field (**a**, **c**) perpendicular and parallel (**b**, **d**) to the sample surface. The temperature gradient: in-plane for (**a**, **b**) and out-of-plane for (**c**, **d**). W and L stand for transverse and longitudinal distances. **e** Illustration of the VNE with electrical detection in open circuit conditions. **f** Resume of emerging effects with respect to our experimental set-up in each configuration (**a**–**d**): OSE (all the stack), ISHE (NiFe, WSe$_2$), (S)VNE (WSe$_2$), ANE (NiFe), ONE (all the stack), PNE (NiFe), RE (NiFe) and SSE (NiFe) stand for ordinary Seebeck effect, inverse spin Hall effect, (spin) valley Nernst effect, anomalous Nernst effect, ordinary Nernst effect, planar Nernst effect, rectification effects (due to the anisotropic magnetoresistance and anomalous Hall effect) and spin Seebeck effect, respectively.

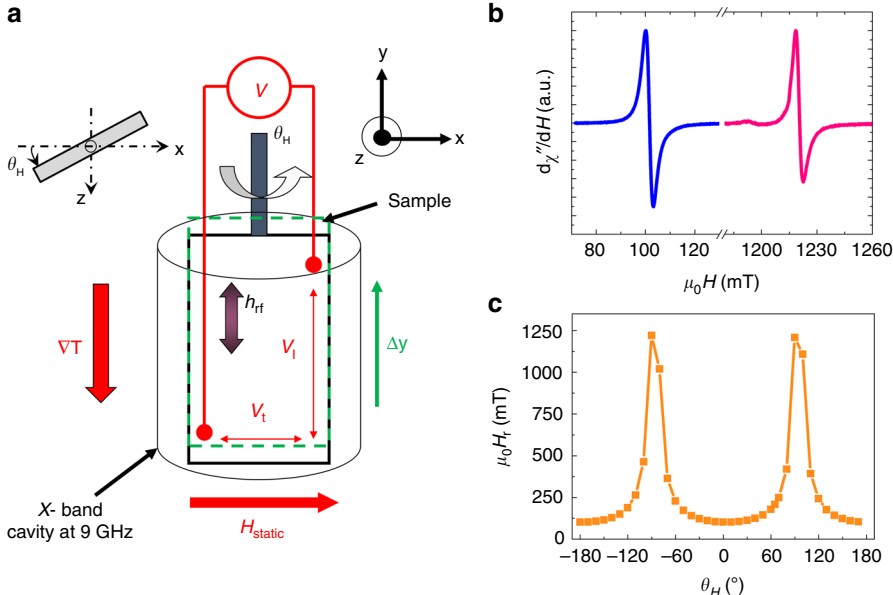

**Fig. 3 FMR and voltage measurements. a** Schematic of the spin-pumping experiment in the FMR cavity where the sample is vertically shifted $\Delta y = 1$ mm away from the cavity center. **b** Typical FMR spectra recorded at $\theta_H = 0°$ and $90°$ and **c** angular dependence of the resonance magnetic field of the stack Al/NiFe/Al/WSe$_2$ (monolayer).

Both voltage components contain a symmetric (s) and asymmetric (a) part with respect to the static magnetic field $H_{static}$ defined as: $V^s_{l,t} = (V_{l,t}(\theta_H) + V_{l,t}(\theta_H + 180))/2$ and $V^a_{l,t} = (V_{l,t}(\theta_H) - V_{l,t}(\theta_H + 180))/2$. $V^s_l$ is the voltage due to the Seebeck effect as a consequence of the off-centered position of the sample in the cavity and the resulting vertical temperature gradient. $V^a_l$ corresponds to the inverse spin Hall effect (ISHE) in the parallel configuration ($\theta_H = 0°$ and $180°$) while $V^a_t$ corresponds to the ordinary Nernst effect, anomalous Nernst effect (ANE), spin Nernst effect and VNE in the perpendicular configuration ($\theta_H = 90°$ and $270°$) (see table in Fig. 2f). Due to time reversal symmetry, spin Nernst effect and VNE are expected to give zero signal. However, in the perpendicular configuration, spin pumping allows to address a single spin/valley population leading to a net transverse voltage as illustrated in Fig. 2e.

**Observation of the valley Nernst effect**. We performed the measurements at room temperature for four different samples: 2 samples with one monolayer of WSe$_2$ (samples A and B), a reference sample without WSe$_2$ (sample C) and one sample with WSe$_2$ multilayers (sample D). We first focus on sample A and detect a symmetric voltage in Fig. 4a demonstrating the large Seebeck effect in WSe$_2$. The rotating sample disturbs the RF cavity settings and modifies the Q factor (proportional to the RF field $h_{rf}$). Here, the Q factor and the Seebeck effect are minimum in the parallel configuration. The Seebeck voltage ($V^s_l$) varies between 10 and 35 μV with a value of 22.6 μV in the perpendicular geometry. If we assume that, at best, 1/10 of the Nb atoms are activated at room temperature, the expected doping level is $10^{11}$ cm$^{-2}$ giving a Seebeck coefficient $S = 500$ μV K$^{-1}$[26,27]. The temperature difference between the two electrical contacts ($\Delta T$) then ranges between 20 and 70 mK depending on the sample orientation. In the perpendicular configuration, the temperature difference is ~40 mK. $V^a = V^a_l + V^a_t$ is reported in Fig. 4c as a function of the applied field for two different angles $\theta_H = 0°$ (parallel configuration) and $\theta_H = 90°$ (perpendicular configuration), $\mu_0 H_r$ being the resonance field. In Fig. 4b, we show the full angular dependence of $V^a$ for $\mu_0 H = \mu_0 H_r$. We

clearly see that the signal is maximum in the perpendicular configuration.

In Fig. 4e, we repeated the same measurements on sample B with the contacts at different positions and found the same behavior but with different signal amplitudes. Moreover, in sample B, $V^a$ is negative at $\theta_H = 90°$ since we reversed the polarity of the contacts ($V+$ and $V-$) with respect to sample A as shown in the insets of Fig. 4b, e. As a comparison, $V^s_l$ in the reference sample C exhibits very low values in the whole angular range demonstrating that the Seebeck effect is negligible in the Al/NiFe/Al/graphene/SiC stack (see Supplementary Note 2). Furthermore, the full angular dependence of the measured voltage in sample C (Fig. 5a) shows two things: (i) zero voltage in the perpendicular configuration, indicating that the ordinary Nernst effect in the full stack as well as the ANE in NiFe are negligible and not detectable. We also note that the ANE in NiFe (sample A) can be estimated as follows: $V_{ANE} \approx \alpha_{ANE} \times \Delta T \times W/L \approx 0.1$ nV with $\alpha_{ANE}$ in NiFe being of the order of 9 nV K$^{-1}$ (ref. [28]), $\Delta T = 70$ mK and $W \approx 0.2$ mm, $L \approx 1.3$ mm being the transverse and longitudinal distances between the electrical contacts. The ANE voltage is indeed negligibly small compared with the measured $V^a$ in the perpendicular configuration; (ii) a cosine dependence corresponding well to the ISHE. We do not discuss here about the origin of this ISHE signal which probably comes from the self-conversion into the NiFe layer.

**Analysis of the valley Nernst effect in WSe$_2$**. We then interpret the angular dependence of $V^a = V^a_l + V^a_t$ in samples A and B (Fig. 4b, e) as the combination of the ISHE proportional to the in-plane component of the spin $s_{//}$ and the VNE proportional to the out-of-plane component of the spin $s_\perp$. We can express the experimental data as: $V^a = V^0_{ISHE} \times s_{//} + V^0_{VNE} \times s_\perp$ where $V^a = V^0_{ISHE}$ for $\theta_H = 0°$ ($s_\perp = 0$) and $V^a = V^0_{VNE}$ for $\theta_H = 90°$ ($s_{//} = 0$). Here we do not consider the ordinary Nernst effect in WSe$_2$ since it is expected to be at least 3 orders of magnitude smaller than the VNE coefficient as discussed in ref. [15]. Moreover, if there exists a longitudinal temperature gradient as described in Fig. 2d, the contribution of the ANE and SSE in NiFe for the

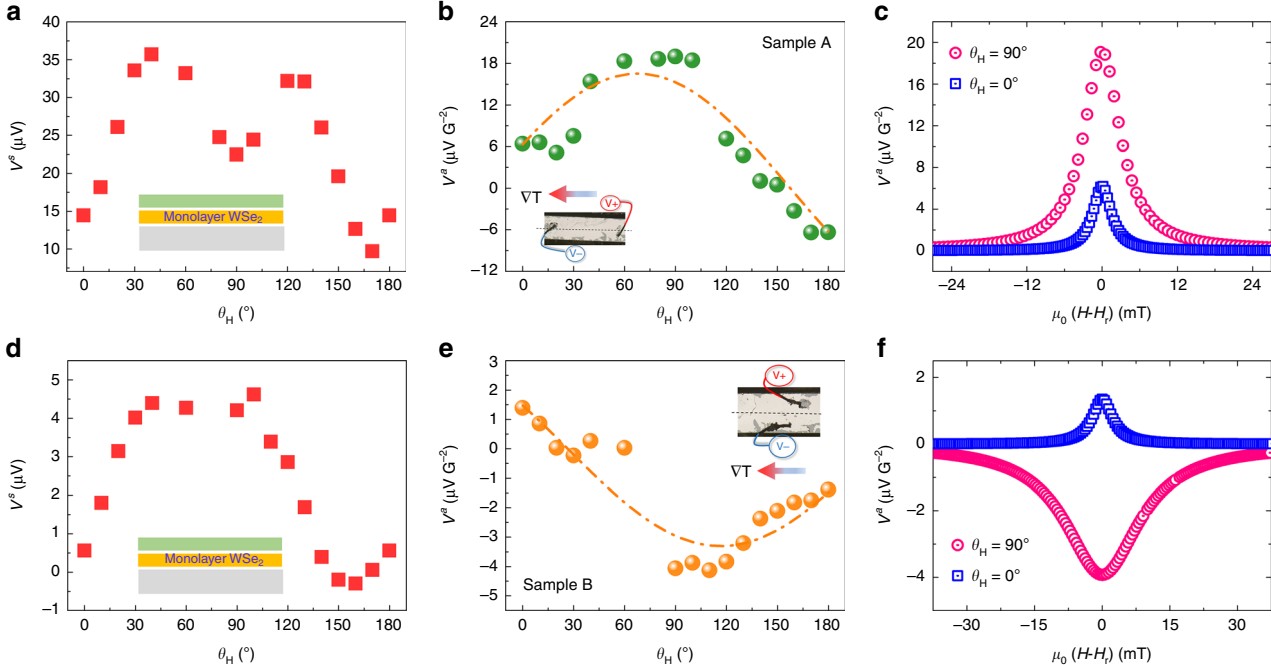

**Fig. 4 VNE in the monolayers WSe$_2$.** Sample A: **a** $V_1^s$ (Seebeck effect) and **b** $V^a = V_1^a + V_t^a$ (ISHE and Nernst-related effects) voltages as a function of the angle $\theta_H$; **c** Parallel ($V^a = V_1^a$, $\theta_H = 0°$, ISHE) and perpendicular ($V^a = V_t^a$, $\theta_H = 90°$, Nernst-related effects) configurations. Sample B: **d** $V_1^s$ and **e** $V^a = V_1^a + V_t^a$ as a function of the angle $\theta_H$; **f** Parallel ($V^a = V_1^a$, $\theta_H = 0°$, ISHE) and perpendicular ($V^a = V_t^a$, $\theta_H = 90°$, Nernst-related effects) configurations. The insets in **b** and **e** are optical images showing longitudinal distances of 1.3 and 0.3 mm between the contacts of the samples A and B, respectively and the dash-dotted lines are guide for the eye. The insets of **a** and **d** show the schematics of the stacks (color code: gray for SiC-graphene substrate, orange for WSe$_2$ and green for Al/NiFe/Al).

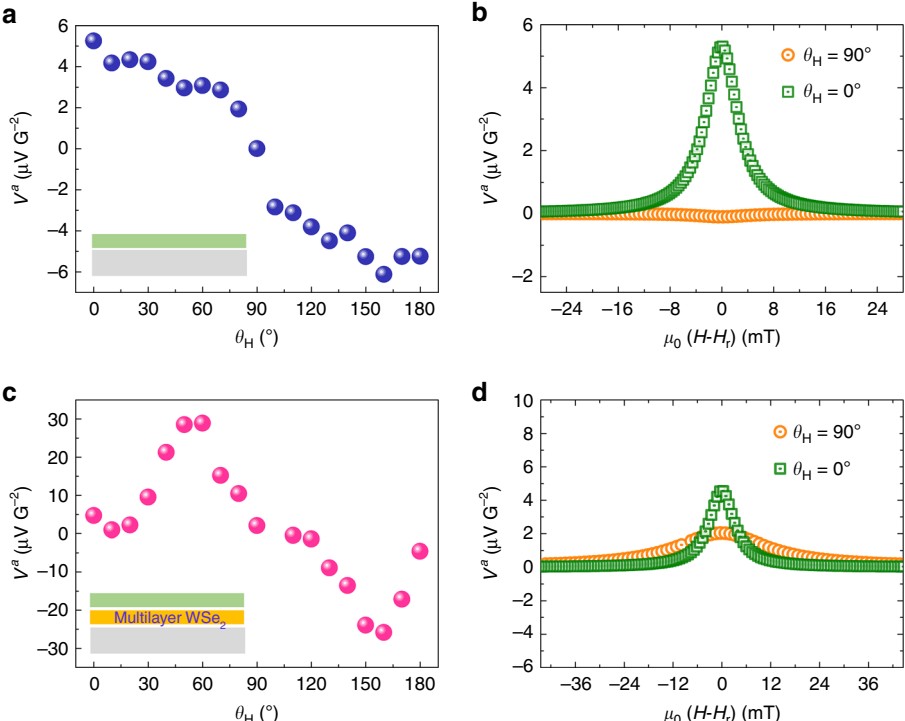

**Fig. 5 VNE in the reference (sample C) and in the multilayer WSe$_2$ (sample D).** Sample C: **a** $V^a = V_1^a + V_t^a$ as a function of the angle $\theta_H$; **b** Parallel ($V^a = V_1^a$, $\theta_H = 0°$, ISHE) and perpendicular ($V^a = V_t^a$, $\theta_H = 90°$, Nernst-related effects) configurations. Sample D: **c** $V^a = V_1^a + V_t^a$ as a function of the angle $\theta_H$; **d** Parallel ($V^a = V_1^a$, $\theta_H = 0°$, ISHE) and perpendicular ($V^a = V_t^a$, $\theta_H = 90°$, Nernst-related effects) configurations. The insets of **a** and **c** show the schematics of the stacks (color code: gray for SiC-graphene substrate, orange for WSe$_2$ and green for Al/NiFe/Al).

in-plane component can also be ignored as demonstrated in NiFe with FMR-SP experiment[29]. We find ($V^0_{\text{ISHE}} = 6.4\ \mu\text{V G}^{-2}$, $V^0_{\text{VNE}} = 18.9\ \mu\text{V G}^{-2}$) for sample A and ($V^0_{\text{ISHE}} = 1.2\ \mu\text{V G}^{-2}$, $V^0_{\text{VNE}} = -4.1\ \mu\text{V G}^{-2}$) for sample B. The VNE coefficient is given by: $\alpha_{\text{VNE}} = V^0_{\text{VNE}}/(R\Delta T) = S V^0_{\text{VNE}}/(R V^s_1)$ where $R$ is the WSe$_2$ transverse resistance between the two electrical contacts. This resistance is measured independently by transferring the same WSe$_2$ film grown on mica onto a SiO$_2$/Si substrate pre-patterned with a transfer length measurement (TLM) arrangement (see Supplementary Note 4). We find the VNE coefficient $\alpha_{\text{VNE}} \approx$ 38.9 pA K$^{-1}$ for sample A and $\approx$ 38.8 pA K$^{-1}$ for sample B. If we define $\alpha_0 = e k_B/2h$ where $e$ is the electron charge, $h$ the Planck constant and $k_B$ the Boltzmann constant, we obtain $\alpha_{\text{VNE}} \approx 0.02\alpha_0$ for one monolayer of WSe$_2$.

## Discussion

In the following, we assume that Nb doping shifts the Fermi level close to the top of the $K^+ \uparrow$ and $K^- \downarrow$ valence bands. We also assume that the intervalley scattering time is longer than the characteristic momentum scattering time in the WSe$_2$ layers. The VNE thus generates a pure transverse valley/spin current. The VNE coefficient vanishes when $K^+ \uparrow$ and $K^- \downarrow$ bands are equally populated by time reversal symmetry. If we assume that FMR-SP in the perpendicular configuration allows to populate only the $K^+ \uparrow$ valley (100% valley polarization), the maximum expected value for the VNE coefficient is $\approx 0.09\alpha_0$ at room temperature when the Fermi level is located between the $K^+ \uparrow$, $K^- \downarrow$ valence band edge[14]. Experimentally, we find $\alpha_{\text{VNE}} \approx 0.02\alpha_0$ which is very close to the predicted maximum value. The experimental value suggests that the Fermi level lies approximately $\Delta E \approx 0.11$ eV above the top of the valence band[14]. The Fermi energy corresponds to a hole density $n_h \approx \frac{g_v m_h^* k_B T}{\pi \hbar^2} \exp(\frac{\Delta E}{k_B T}) = 1.3 \times 10^{11}$ cm$^{-2}$ (where $g_v$, $m_h^*$, $T$, $k_B$, $\hbar$ denote valley degeneracy, effective hole mass, temperature, Boltzmann and reduced Planck constants, respectively), corroborating well with the hole doping range in our WSe$_2$. Other features related to the VNE such as rf-power quadratic dependence and spatial-shift induced sign change of the voltage ($V^a$ at $\theta_H = 90°$) are given in Supplementary Note 3. Our interpretation for the observation of VNE is also supported by the fact that we detect no voltage in the perpendicular configuration in the reference sample C (Fig. 5a, b). Moreover, we performed the same FMR-SP measurements on the multilayer WSe$_2$ as shown in Fig. 5c, d. In this case, the perpendicular voltage is much weaker than the one of the monolayer WSe$_2$ and comparable to the in-plane signal. This is also true when we convert the voltages into currents by normalizing with the resistance measured in open-circuit condition: 0.18, 0.00 and 0.06 $\mu$AG$^{-2}$ for samples A, C and D, respectively. In particular, the maximum voltage is no longer located at $\theta_H = 90°$. This angular dependence of the voltage is still in qualitative agreement with the observation of the valley Nernst effect. First, when increasing the number of layers, WSe$_2$ becomes an indirect bandgap semiconductor with the top of the valence band at the $\Gamma$ point. As a consequence, when increasing the number of layers, we expect a sharp decrease of the valley Nernst effect in the $K$ valleys as observed experimentally. Second, the angular dependence of the transverse voltage $V^a$ of the monolayer, which shows a maximum when the spin of holes is pinned to the out-of-plane direction, results from the imbalance of the valley populations. Here, any signal arising from the Rashba spin–orbit coupling (induced by the asymmetric environment of WSe$_2$ or induced by the proximity effect in graphene[30]) is neglected. In particular, the recently discussed spin-type Berry curvature[31] is orders of magnitude smaller than the intrinsic Berry curvature at the valence band edges of WSe$_2$ monolayer. Finally, we note that the full angular dependence of the FMR-SP measurements in the

case of the multilayer WSe$_2$ (Fig. 5c) needs further investigation in order to get insight into its behavior, in particular the appearance of the two maxima with opposite signs. We anticipate that the direct-to-indirect bandgap transition and Ising-Rashba spin–orbit effects[32] associated with the restored inversion symmetry in the multilayer would be appropriate arguments to interpret the full angular dependence.

In conclusion, by applying a macroscopic temperature gradient and by addressing a single $K$ valley, we could demonstrate the large valley Nernst effect in monolayer WSe$_2$. For this, we have grown high-quality WSe$_2$ monolayers on graphene by van der Waals epitaxy over large areas. At the ferromagnetic resonance of the NiFe film in contact with WSe$_2$, we could both generate a temperature gradient in WSe$_2$ at the millimeter scale and a valley polarization by spin pumping. This original technique allowed us to give an estimation of the valley Nernst coefficient that agrees well with the predicted value for one monolayer of WSe$_2$. The valley Nernst effect could be used as an alternative way to generate valley currents for valley-related physics.

## Methods

**Sample growth and characterization**. The graphene/SiC substrate is first outgased at 750 °C during one hour in the MBE chamber with a base pressure in the low $10^{-10}$ mbar range. It is then kept at 500 °C during the co-deposition of W (0.01875 Ås$^{-1}$), Se ($10^{-6}$ mbar) and Nb (0.01875 Ås$^{-1}$). W and Nb are evaporated thanks to e-gun evaporators and Se with a Knudsen cell. The Nb flux is pulsed in order to reach the right concentration of 0.1%. At the end of the growth, the whole film is annealed at 750 °C during 15 minutes under Se flux to improve the crystal quality[33]. We obtain continuous and uniform WSe$_2$ films on $5 \times 5$ mm$^2$ graphene/SiC substrate.

X-ray diffraction analysis was performed with a SmartLab Rigaku diffractometer equipped with a Copper Rotating anode beam tube (K$_\alpha$ = 1.54 Å) and operated at 45 kV and 200 mA. A parabolic mirror and a parallel in-plane collimator of 0.5 °C of resolution was used in the primary optics and a second parallel collimator was used in the secondary side. A K$_\beta$ filter was used for all the measurements.

Raman measurements were done with a Horiba Raman set-up using a laser excitation source of 632 nm with spot size of 0.5 $\mu$m. The signal was collected by choosing a 1800 grooves/mm grating.

**WSe$_2$ transfer**. We adopted a wet transfer process that is described in details in ref. [34] for the transfer of WSe$_2$ grown on mica onto SiO$_2$/Si substrate.

**Electrical contacts**. The Ti/Pt contacts, which were pre-patterned on SiO$_2$/Si substrate by lithography and etching followed by metal deposition, have thicknesses of 5 nm/10 nm, respectively. See Supplementary Note 4 for the distribution of the contact distance.

## Data availability

The data that support the findings of this study are available from the corresponding author upon reasonable request.

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

## Acknowledgements

The authors acknowledge the financial support from the ANR projects MoS2Valley-Control and MAGICVALLEY. The LANEF framework (No. ANR-10-LABX-51-01) is also acknowledged for its support with mutualized infrastructure. One of us (BG) thanks ANR J2D for financial support (nc-AFM characterizations). We thank A. Michon for his help in the preparation of Graphene/SiC subtrate, Y. Genuist for his help in the growth of metallic stacks, N. Mollard for preparation of TEM samples and V. Mareau, L. Gonon for their assistance in Raman measurements.

## Author contributions

M.T.D. performed the spin pumping measurements with the help of S.G. and V.M. M.T.D. and M.J. analyzed and interpreted the spin pumping. C.V., C.B., M.T.D., T.G. and M.J. performed the transfer and electrical measurements. M.T.D. and C.V. grew the $WSe_2$ layers and carried out the Raman measurements. T.J. and B.H. performed the growth of graphene on SiC substrate. A.M. carried out and analyzed the X-ray diffraction data. B.G. performed the atomic force microscopy characterization. H.O. made the transmission electron microscopy observations. M.T.D. and M.J. wrote the paper with comments of all co-authors. M.J. initiated the study and supervised the project.

## Competing interests

The authors declare no competing interests.
