## [Peer Review File · Nature Communications]

REVIEWERS' COMMENTS:

Reviewer #2 (Remarks to the Author):

I am satisfied with the revision. Although the style of the manuscript should be changed from Letters to Articles, I can recommend the publication of this paper in Nature Communications.

Additional comments to be addressed before acceptance:

(1) The composition of NiFe is not shown.

(2) Too many abbreviations are used. To improve the readability, "BC", "SNE", "ONE", and "RE" should be spelled out, which are used only a few times. "OSE" and "PNE" are not defined in the main text, and should be spelled out.

Point-by-point response to reviewers

REVIEWERS' COMMENTS:

Reviewer #2 (Remarks to the Author):

I am satisfied with the revision. Although the style of the manuscript should be changed from Letters to Articles, I can recommend the publication of this paper in Nature Communications.

Additional comments to be addressed before acceptance:

(1) The composition of NiFe is not shown.

(2) Too many abbreviations are used. To improve the readability, "BC", "SNE", "ONE", and "RE" should be spelled out, which are used only a few times. "OSE" and "PNE" are not defined in the main text, and should be spelled out.

Reply: We thank the reviewer for these comments. We have included the composition of the permalloy (Ni₈₁Fe₁₉) in the revised manuscript. We have also spelt out the abbreviations which are occasionally used in the manuscript and defined the "OSE" in the main text. Note that all the abbreviations are defined (and reminded) in the legend of Figure 2.